# The Zebra Mussel (*Dreissena polymorpha*) as a Model Organism for Ecotoxicological Studies: A Prior ^1^H NMR Spectrum Interpretation of a Whole Body Extract for Metabolism Monitoring

**DOI:** 10.3390/metabo10060256

**Published:** 2020-06-18

**Authors:** Sophie Martine Prud’homme, Younes Mohamed Ismail Hani, Neil Cox, Guy Lippens, Jean-Marc Nuzillard, Alain Geffard

**Affiliations:** 1Stress Environnementaux et Biosurveillance des milieux aquatiques, Université Reims Champagne Ardenne, UMR-I 02 SEBIO, 51687 Reims, France; younes.hani@u-bordeaux.fr; 2LIEC Lab, Université de Lorraine, CNRS, F-57000 Metz, France; 3Université de Bordeaux, UMR EPOC 5805, équipe Ecotoxicologie aquatique, Place du Dr Peyneau, 33120 Arcachon, France; 4Toulouse Biotechnology Institue (TBI), Université de Toulouse, CNRS, INRA, INSA de Toulouse, 135 avenue de Rangueil, 31077 Toulouse CEDEX 04, France; ncox@insa-toulouse.fr (N.C.); glippens@insa-toulouse.fr (G.L.); 5Institut de Chimie Moléculaire, Université de Reims Champagne Ardenne, UMR CNRS 7312 ICMR, 51097 Reims, France; jm.nuzillard@univ-reims.fr

**Keywords:** *Dreissena polymorpha*, zebra mussel, NMR, metabolomics, annotation, ecotoxicology

## Abstract

The zebra mussel (*Dreissena polymorpha*) represents a useful reference organism for the ecotoxicological study of inland waters, especially for the characterization of the disturbances induced by human activities. A nuclear magnetic resonance (NMR)-based metabolomic approach was developed on this species. The investigation of its informative potential required the prior interpretation of a reference ^1^H NMR spectrum of a lipid-free zebra mussel extract. After the extraction of polar metabolites from a pool of whole-body *D. polymorpha* powder, the resulting highly complex 1D ^1^H NMR spectrum was interpreted and annotated through the analysis of the corresponding 2D homonuclear and heteronuclear NMR spectra. The spectrum interpretation was completed and validated by means of sample spiking with 24 commercial compounds. Among the 238 detected ^1^H signals, 53% were assigned, resulting in the identification of 37 metabolites with certainty or high confidence, while 5 metabolites were only putatively identified. The description of such a reference spectrum and its annotation are expected to speed up future analyses and interpretations of NMR-based metabolomic studies on *D. polymorpha* and to facilitate further explorations of the impact of environmental changes on its physiological state, more particularly in the context of large-scale ecological and ecotoxicological studies.

## 1. Introduction

Freshwater aquatic systems are contaminated by a broad variety of pollutants due to human activities (agriculture, industries, urban effluents, transport activities, etc.) [1,2,3]. Determining the consequences of these contaminants on aquatic biota is a major challenge for ecotoxicologists, and ongoing assessments of aquatic ecosystems’ states rely on the development of sensitive and high-throughput techniques for the detection of sub-lethal responses in indigenous organisms [4]. 

The zebra mussel, *Dreissena polymorpha*, is a reference organism for inland water ecotoxicological studies. This invasive species has a wide distribution in European and North American lentic and lotic water ecosystems. As a not endangered and continuously available species which has a sessile behavior and an adequate body size, it is relatively easy to sample. In addition, its long lifespan and high filtration rate favor its uptake of and thus exposure to environmental pollutants. The zebra mussel hence is a useful sentinel organism that responds quickly to pollution changes and thereby could give access to a straightforward identification of their negative physiological effects [5]. A large set of biomarkers (targeted biochemical and/or physiological changes characteristic of contaminant exposure) has been developed for this species (e.g., [6,7,8,9]), and has been exploited in numerous laboratory and field studies [9,10,11,12,13]. These biomarkers are thought to reflect the initial response of zebra mussels to environmental contamination. However, they often lack representativeness of the physiological state of the studied individuals and do not allow understanding the molecular events leading to observed perturbations and responses. Hence, the need is often raised for the development and integration of more global and untargeted approaches, considering the physiology on a molecular scale in a global way [5,14,15,16].

Initially developed in the clinical and biomedical fields, metabolomics has gained interest in ecotoxicology during the last decade [17,18]. The aim of this approach is the characterization of a set of metabolites, or metabolome, which is the collection of low-molecular weight molecules (typically of mass lower than 1500 Da) present in a cell, tissue or organism at a given time. Being the result of gene expression and the end-product of the metabolism, the metabolome is considered representative of the physiological state of an organism [19]. In an ecotoxicological context, metabolomic studies offer the potential to identify changes in environmental conditions in a biomonitoring context, and to understand the consequences of such changes on the organism’s physiology. If untargeted, this approach can also reveal unexpected perturbations of physiological functions, leading to new hypotheses for the environmental risk assessment. In addition, metabolomic approaches have the great advantage of being applicable to any species without any prior knowledge of its genome. Indeed, the development of other “-omic” approaches (proteomic, transcriptomic, etc.) in ecotoxicological studies is mainly limited by the lack of genomic data on most of the environmentally relevant species [17,18]. Metabolomics has been involved in several studies related to the impact of contaminants on numerous aquatic organisms [20], including marine bivalve species in laboratories (e.g., [21,22,23]) or in the field (e.g., [4,24,25,26,27]).

Metabolomic studies rely on two main analytical methods: mass spectrometry (MS) and nuclear magnetic resonance (NMR) spectroscopy. Although suffering from a lower sensitivity, NMR-based metabolomics approaches enable a non-targeted analysis of the most abundant metabolites of an organism or tissue extract, in a highly accurate, reproducible and quantitative way. NMR spectra are virtually independent of the operator and spectrometer, ensuring a high reliability to the recorded data [28]. In addition, sample preparation requires minimal effort and time, and spectral data acquisition of a single proton 1D spectrum is a matter of minutes, making the comparison of a large set of samples possible [17,28]. NMR is thus adapted to ecotoxicological biomonitoring field studies dealing with large sets of sites, and to standard surveys devoted to the detection of potentially impacting environmental conditions. However, whatever the employed analytical method, signal assignment remains one of the main challenges of metabolomics [17,29,30]. In most NMR-based ecotoxicological studies, spectral peak assignment is solely based on chemical shift values [22,23,24,25,26,27], and only rarely relies on additional experimental data like heteronuclear ^1^H-^13^C chemical shift correlations. However, chemical shift values may vary according to the sample pH, to the presence of other metabolites in the sample, and of course, the internal standard used for spectrum referencing (most of the time, either sodium trimethylsilylpropanesulfonate or sodium 3-(trimethylsilyl)-2,2,3,3-tetradeuteropropionate). As a consequence, a positive match in databases does not necessarily imply an accurate metabolite identification. In addition, many metabolites lack complete 1D and 2D NMR reference data in public databases [29,30,31]. As a result, many signals can remain unassigned, and some may even be inaccurately assigned to a metabolite, which limits the data interpretation in NMR-based metabolomic studies and therefore the accuracy of the conclusions drawn from them. Given that the majority of metabolomic approaches in ecotoxicology aim at identifying specific biomarkers of contaminant exposure or environmental changes, providing thorough and reliable annotations for environmentally relevant species is of key importance. In addition, spectral data recorded during the course of such studies are most of times only available as un-FAIR (Findable, Accessible, Interoperable, Reusable) [32] low-resolution drawings, making almost impossible any further exploitation and comparison by interested readers, even though solutions have been proposed for the storage of raw and interpreted NMR data [33,34]. The lack of exploitable accurately annotated reference metabolomic spectra strongly limits the co-construction of a common metabolome interpretation, as every new scientific team that want to use NMR data have to start again with their own (putative) interpretation of their own spectra.

In an interesting initial feasibility study, Watanabe et al. demonstrated the high potential of the freshwater species *D. polymorpha*’s metabolomics for a water quality evaluation [4]. Their work evidenced metabolic profile variations in the ^1^H NMR spectra of individuals from different sampling stations undergoing anthropic pressure. However, as the majority of NMR-based metabolomic ecotoxicological studies, their spectrum interpretation relied on a putative signal assignment, based on a comparison of proton and carbon chemical shifts extracted from 1D ^1^H and 2D heteronuclear ^1^H-^13^C spectra with databases. In addition, the lack of signal shape description in their annotation and the low resolution of their representative annotated spectrum unfortunately make it in practice unusable for a reader who would want to start from their putative annotations.

The goal of the present work is to facilitate and increase the reliability of future work on the *D. polymorpha* metabolome by any scientist. Considering the lack of consistent and usable annotation for *D. polymorpha,* this required the thorough and cautious interpretation of a ^1^H NMR spectrum representative of the metabolic profile of *D. polymorpha,* in order (1) to thoroughly and reliably assess the informative potential of ^1^H-NMR-based metabolomics approaches by reporting the nature of detected and identified metabolites and (2) to provide a reliable and usable annotated *D. polymorpha* spectrum facilitating future metabolomic approaches, and favoring better accuracy to biological interpretations on this model species. The detailed annotation of a ^1^H-NMR representative spectrum was obtained through the exploitation of 2D ^1^H-^1^H, ^1^H-^13^C and ^1^H-^31^P NMR spectra, associated with sample spiking for several selected metabolites. This study produces thereby a detailed annotated high-resolution reference ^1^H spectrum, provided as an open-access Appendix A, and the association of 53% of the identified signals with the corresponding metabolite names.

## 2. Results

A representative ^1^H NMR spectrum of the *D. polymorpha* whole-body polar extract is presented in Figure 1. Only the most representative signals for each identified metabolite are indicated. An annotated high-resolution ^1^H spectrum is also available online as an open format JCAMP-DX [35] (see Appendix A section). The 1D and 2D JRES ^1^H spectra of the *D. polymorpha* whole-body polar metabolome allowed identifying 238 signals of various multiplicity (from singlets to highly complex multiplets). The detailed list of these signals, including multiplicity, coupling constants, ^1^H and ^13^C chemical shifts and the nature of the corresponding metabolite is provided in Appendix A.

### 2.1. 1D ^1^H Reference Spectrum Annotation

After the spectrum analysis, 17 metabolites were identified with certainty (level 1), and 20 metabolites were putatively identified with high (level 2+) confidence, for a total of 37 identified metabolites (Table 1, Appendix A). In addition, 20 unknown groups of signals or well-defined singlets were identified and named “Unknown M1” to “Unknown M23” (Appendix A). A total of 113 signals were assigned to a specific metabolite (levels 1 to 2+) and 39 signals were assigned to unknown metabolites. Among all the detected signals, 38% (89 peaks) remained unassigned to any known or unknown metabolites, due to the lack of concordant information obtained for them.

The *D. polymorpha* metabolome during the resting period is dominated by an intense unknown singlet (M5, 78), putrescine (33 and 77), trimethylamine (71), succinate (55) and betaine (91) (Figure 1, Appendix A). Smaller signals are due to the low-concentration polar metabolites belonging to several functional classes: 15 proteinogenic amino acids, 7 compounds with amine and amide functional groups, 3 carboxylic acids, 7 compounds in the nucleotide or nucleoside family, 2 coenzymes and 5 carbohydrates (Table 1).

### 2.2. Metabolites Assigned as Unknown

Among the 23 unidentified metabolites (Appendix A), the M1 signals (37, 56 and 57) are relatively abundant and well-defined (see Figure 1B), and were putatively assigned to γ-aminobutyric acid (GABA, Peakforest, PFs000226). However, it remains unclear why we observe two identical triplets (56 and 57) correlated with the signal 37 instead of a single one as in the reference database. This may be due to the resonance of a closely related metabolite, co-resonating with GABA at the chemical shift of the signal 56, and with a slightly shifted identical triplet at 2.45 ppm (56 or 57). The only metabolite referenced in the databases that could correspond to the M10 singlet (174) was allantoin (PFs000436). However, as we did not identify any correlation with the ^13^C peaks, we could not confirm this annotation. M22 and M23 correspond to two singlets (85 and 86) that both correlate with a carbon at 56.7 ppm. According to this information, these two singlets probably correspond to phosphocholine (HMDB01565) and glycerophosphocholine (HMDB00086). According to their relative chemical shift, we can suggest that M22 corresponds to phosphocholine and M23 corresponds to glycerophosphocholine, but due to the lack of homonuclear correlation for these two signals, we do not have enough arguments to confirm that annotation, which remains putative. In the same way, the only correspondence found for signals 241, 240 and 237 was nicotinamide ribotide (NMN; HMDB00229). However, as no homonuclear correlations have been detected for these low-intensity signals, we could not firmly confirm their belonging to a metabolite. These five putative metabolite assignments were thus considered with moderate confidence (levels 2−).

The M11 and M12 groups of signals produce characteristic pairs of doublets into the 5.5 ppm region (signals 178–179 and 180–181, respectively, see Figure 1E). This kind of characteristic equal pairs of doublets has been identified in the same region when spiking glucose 1-phosphate (Appendix A). Similar pairs of doublets can also be found in the uridine diphosphoglucose reference spectrum (PFs000652). In addition, a correlation with a ^31^P peak at 1.70 ppm has been identified in the HSQC ^1^H-^31^P spectrum for signals 178 and 179. No phosphorus correlation has been identified for signals 180 and 181, but as they are threefold less intense than signals 178 and 179, they might fall below the detection threshold. As sugar phosphate and sugar phosphate-containing metabolites such as nucleotide sugar are not widely referenced in the databases, M11 and M12 were referred to as “sugar 1-phosphate” (level 3 in the identification quality scale).

Signals 185, 186, 187, 191, 192 and 197 belonging respectively to M14, M13, M15, M16, M17 and M18 are all located in the resonance region of ribose C2 of nucleotides and nucleosides (see Figure 1E). Signals 186 and 187 are located at the same chemical shift as UDP and UTP. However, spiking of UDP and UTP yielded peaks absent in our reference spectrum, and hence, UDP and UTP are considered as undetectable in this spectrum. Signals 186 and 187 may be due to the resonance of other uridine-phosphate containing molecules. However, nucleotides, nucleosides and nucleosides containing metabolites are poorly referenced in the databases, and as without direct spiking, misidentifications could be easily made, and the annotation of these unknown metabolites was limited to a level 3 as nucleotide/nucleoside compound class (level 3 of identification).

The ^1^H and ^13^C chemical shift of the M2 singlet (43) could correspond to the acetyl group of several metabolites in the HMDB database. In addition, there was no correlation signal in ^1^H-^31^P HSQC for this singlet. M1 thus probably belongs to the class of acetylated compounds, but we could not go further in the assignment (level 3). On the contrary, despite several homonuclear and carbon correlations that could be identified for peaks from M4, M7 and M8, no correspondence in the databases was found, suggesting that these metabolites are not referenced yet. However, phosphorus correlations were identified in the ^1^H-^31^P HSQC spectrum for signals 90, 152 and 153, revealing that these three metabolites are phosphorylated compounds (level 3 of identification) (Appendix A).

### 2.3. Signal Superposition in the ^1^H NMR Spectrum

The JRES spectrum revealed that in several regions, numerous peaks are located at the same place. Signal superposition is evidenced in the publicly available high-resolution annotated spectrum (see Appendix A section). The spectral region between 3.6 and 4.4 ppm is a particularly crowded region, with many correlated resonances in the 2D COSY spectrum (Figure 2). Carbohydrates and the sugar part of nucleotides and nucleosides (and more complex nucleotide/nucleoside-based metabolites such as NAD/NADH) and protons of α-carbons of amino acids all contain protons that resonate in this region. This resulting complex peak cluster, to which many metabolites participate, is therefore the resultant of all metabolites, even though some signals of abundant metabolites can be recognized (e.g., AMP peaks 147 and 148). In the same way, the spectral region from 1.80 to 1.94 ppm is highly complex, with several amino acids side chain resonances. For this reason, signal positions and assignments were not detailed for these two particular regions of the annotated high-resolution spectrum (see Appendix A section), while detected peaks on the JRES spectrum are indexed in the peak list (Appendix A). It is to note that lactate and threonine doublets resonating at 1.33 ppm (peaks 21 and 22, Figure 1A) are nearly not distinguishable in the 1D ^1^H spectrum, even if both are identified in the JRES spectrum.

## 3. Discussion

*D. polymorpha* is a reference organism in ecotoxicology, used as a bioindicator of the chemical and biological contamination state of freshwaters [5,9,11,36], and for the characterization of contaminant impacts on its physiology [7,13,37,38]. A growing interest for this species motivates the development of new approaches for the evaluation of its physiological state. Our work led to the reliable characterization of the whole-body metabolome of this species.

By exploiting the homonuclear and heteronuclear 2D NMR spectra, we were able to thoroughly explore the complex spectrum resulting from the ^1^H resonances of all small polar free molecules extracted from dry *D. polymorpha* whole-body powder. The 2D JRES spectrum allowed the determination of peak multiplicity and to group all individual peaks in the 1D ^1^H spectrum according to the multiplet signal they belong to. A clear view of signal superposition in the spectrum is also provided. The 2D ^1^H-^1^H COSY and TOCSY spectra allowed grouping signals as resonances within a given metabolite. This process enabled us to check the chemical shift correspondences with the reference databases of all the peaks of a given metabolite simultaneously, and not only on a peak by peak basis, as it is usually performed when studies are limited to 1D ^1^H NMR approaches. This approach was taken a step further by the additional use of a ^1^H-^31^P HSQC spectrum, allowing the identification of phosphorylated compounds among the unknown metabolites. The spiking of reference metabolites directly into the sample confirmed, discriminated or invalidated the putative signal assignment based on database mining. This method appeared particularly effective to deal with closely related groups of metabolites like, for example, nucleotides with different levels of phosphorylation, that produce very similar and sometimes even coinciding peaks. It is also highly useful for small metabolites with nuclei resonating as isolated singlets, for which it is impossible to compare anything else than the peak position. This was for example the case when we spiked sarcosine. Searching databases for singlets 66 or 67 suggested the possible presence of sarcosine, with a good fit of the singlet 66 in both dimensions of the ^1^H and ^13^C chemical shifts. However, spiking the sample with sarcosine revealed that none of these singlets were actually due to sarcosine, and that this metabolite was under the detection level in our reference spectra.

Efforts made at characterizing the ^1^H NMR spectrum of the *D. polymorpha* whole body led to the assignment of 53% of detected signals to specific metabolites, with confidence levels ranging from certainty through metabolite spiking, to putative, based only on comparison to the reference databases. Among all detected signals, 12.5% were assigned to “unknown metabolite”, for whose no assignment or only compound class were identified. As a result, 37 metabolites were identified with certainty or good confidence, and additional 3 were putatively identified. Additional spiking experiments would be necessary to firmly confirm the assignation of these three metabolites. The only NMR-based metabolomic study on *D. polymorpha* to date has been conducted by Watanabe et al. [4]. In their work, they identified 35 putative metabolites with limited reliability in the metabolome of zebra mussels during the resting period, based on the ^1^H and ^13^C chemical shift values derived from ^1^H-^13^C HSQC and by their comparison with databases. The higher number of identified metabolites in the present study could be due to higher assignment efforts, but also to the used methodology, where we caged mussels from the same population in diverse sites, ensuring mixing diverse metabolic profiles in our representative samples. Interestingly, although Watanabe et al. [4] identified several metabolites that we also identified in our reference spectra, the overlap was far from perfect, with 10 metabolites of that study unidentified in our spectra, and 14 metabolites identified in our reference metabolome not identified by Watanabe et al. [4]. These discrepancies could be either due to variations in the abundance of some metabolites between the two studies or to different levels of stringency in the assignment process from the databases. For example, contrarily to the pre-cited authors, we considered that we could not attribute the pair of doublets 178–179 (M11, sugar 1-phosphate 1) to UDP-N-acetylglucosamine considering that sugar-1-phosphates are too poorly referenced in databases. In addition, given their proximity, both pairs of the doublets 178–179 and 180–181 could correspond to this metabolite. We thus chose to propose only a level 3 confidence level for this identification. Overall, any further comparison between our study and theirs is difficult, as they do not detail the shape of each assigned signal, and only a tiny low-resolution spectrum is provided with the article, like in most of ecotoxicological metabolomic studies [33]. Our high-resolution annotated spectrum thereby should allow readers to compare their own spectra with ours (see Section 3.3).

### 3.1. Informative Potential of NMR Based Metabolomics in D. polymorpha

Several compound classes are represented among the 42 identified metabolites. The *D. polymorpha* metabolome is dominated by five metabolites: unknown M5, putrescine, trimethylamine, succinate and betaine (Figure 1). Among the lower intensity signals, 15 proteinogenic amino acids, 7 other amine and amide compounds, 3 carboxylic acids, 7 nucleotide or nucleoside compounds, 2 coenzymes and 3 carbohydrates were identified (Table 1). The abundance of these metabolites provides an insight into the various metabolic pathways and physiological functions.

Energy storage metabolites (glycogen, glucose, maltose), metabolites involved in the citrate cycle (succinic acid) and glycolysis/gluconeogenesis (lactic acid) and direct energy source (ATP) provide information about the energetic state of the organisms. 

The nicotinamide dinucleotide in its oxidized form (βNAD+) and reduced form (NADH) is a cofactor central to the metabolism. Both forms are identifiable at a low level but as well-defined peaks in the reference spectrum, offering the potential to estimate the NAD+/NADH ratio (e.g., with peaks 239 and 231), reflecting the balance between the mitochondrial NADH supply and demand and providing a proxy to estimate the mitochondrial redox state [39]. In addition, precursors of the three NAD biosynthesis pathways are identified in the spectrum: tryptophan, nicotinic acid (niacin) and nicotinamide ribotide (NMN), providing a global view of pathways necessary to maintain the cellular NAD(+/H) pool [40].

Apart from their proteogenic functions, amino acids have important roles in the metabolism of bivalves such as osmoregulation, immunity or energy metabolism. An alteration in amino acids levels when exposed to different environmental contaminants has been suggested to be one of the primary responses of mussels to different stressors [41]. Alanine is notably one of the most abundant proteinogenic amino acids in our reference spectrum. This aliphatic amino acid can form pyruvate via the alanine dehydrogenase activity, entering the citrate cycle and gluconeogenesis [42]. Thus, alanine participates to the energy potential of cells. As mentioned, amino acids act as organic osmolytes in freshwater bivalves [43]. In addition, a lot of detected amine and amide compounds are involved in osmolytic regulation in marine bivalves, such as trimethylamine, dimethylamine, betaine and hypotaurine [44,45]. Arginine, which is also one of the most abundant proteinogenic amino acids in our reference spectrum, is the final step of the urea cycle (or ornithine cycle), forming urea and ornithine due to the action of L-Arginine amidinohydrolase [42]. Aspartate also participates to this cycle, entering it during the argininosuccinate formation step. Unfortunately, we cannot identify ornithine, as its main peaks resonate in very crowded regions of the spectrum. Allantoin was also putatively identified. This metabolite is formed during the purine metabolism, being the product of uric acid oxidation, and leading to ammonia excretion [42]. The identified metabolites thus provide arguments to estimate the state of these two ways of ammonia excretion. 

Putrescine takes an important place in the *D. polymorpha* metabolome. This polyamine is derived from ornithine, and is thus produced by the urea cycle. It is the precursor of both spermidine and γ-aminobutyric acid [42], whereby the latter was putatively identified in our spectrum. Interestingly, it was shown to be involved in the response to oxidative stress induced by cadmium exposure in *Mytillus galloprovincialis*, suggesting a potential role in oxidative stress response [46], which can be interesting in the context of ecotoxicological studies. Gamma-aminobutyric acid (GABA) is another biogenic amine, which can be synthesized from putrescine as mentioned earlier, or from glutamate. It is well-described for its inhibitory neurotransmitter activity, but is also involved in immune response homeostasis in pacific oyster [47] and in adaptive strategies to anoxic or hypoxic conditions in the pen shell *Atrina rigida* [48]. Its levels decrease in the freshwater mussel *Elliptio complanata* upon exposure to neurochemicals [49]. Its abundance could thus provide information about the state of GABAergic signaling, with a potential implication for the immune response state and oxygen supply in the organism. The amino acid tyrosine, which resonates as low-intensity but well-identifiable peaks in our spectrum, is also involved in neurotransmission, through the dopaminergic system, as a substrate precursor for the synthesis of catecholamines [50].

Globally, the ^1^H NMR approach in D*. polymorpha* metabolic studies allows the semi-quantitative identification of key metabolic compounds, representative of major physiological functions such as energy storage and metabolism, osmolytic regulation, excretion and neurotransmission. These metabolites could represent interesting biomarkers of metabolic impairments. However, further studies are necessary to validate the link between metabolites’ modulation and physiological effects. 

### 3.2. Dealing with Metabolome Spectrum Complexity 

The 2D JRES spectrum revealed that in several cases, more than one metabolite was contributing to a given peak observable on the 1D spectrum. This cannot be avoided, since series of metabolites share the same characteristic structure, leading to closely resonating signals. However, by providing a precise list of observable peaks and an annotated spectrum, our work allows the reader to weigh his interpretations through identifying whether several metabolites contribute to a peak or a peak cluster. As an example, we showed the complexity of the 3.6 to 4.4 ppm region. This does not mean that this region is deprived of information. Interestingly, most of the correlations to a ^31^P nucleus identified on the ^1^H-^31^P HSQC spectrum are located in this complex region. Given the major metabolic and energetic functions of phosphorylated compounds in the organism’s physiology, the informative interest of this region is even more real. Two-dimensional approaches coupling proton and phosphorus signals could help to further exploit this information, but this is limited at this time by the lack of ^1^H-^31^P HSQC reference spectra for phosphorylated metabolites in the available databases. A comparison of the proton spectra from several samples still could allow identifying variations in the global profile of this region that would characterize metabolic changes. However, it would be adventurous to attribute such variation to a specific metabolite or group of metabolites. To avoid this problem of superposed signals, it has been suggested to perform metabolomic studies on JRES spectra rather than on plain 1D spectra [51]. This would greatly increase the acquisition time of spectra, limiting the feasibility in large-scale studies, such as multi-site approaches. This is, however, an approach to consider in the context of mechanistic studies, aiming at characterizing precisely metabolite changes in specific conditions. 

For a metabolomic data analysis, spectra are often divided in small equal parts of a few hundreds of ppm, called buckets, before a multivariate statistical exploration of the data. Another possibility is to perform an “intelligent bucketing” allowing to optimize each bucket size and position to exactly match one resonance peak, providing some chemical meaning to the buckets [52,53]. In such a dataset, each bucket content constitutes a variable. Given that a single metabolite can produce several resonance peaks, the more complex the metabolite is (i.e., producing a complex fingerprint ^1^H NMR spectra), the higher the number of variables that will co-vary in the dataset due to the abundance change in this metabolite. Abundance variation of complex metabolites will thus have more impact on sample separation in multivariate analyses than metabolites producing less peaks (i.e., less variables), which could induce a bias. The precise annotation provided here allows the user to merge all buckets belonging to a same multiplet signal, and even belonging to a same metabolite before the multivariate analysis. This results in a reduction in the number of variables, with ideally a single variable per metabolite. This approach is facilitated in the free online statistical analysis platform Biostatflow [54]. It proposes to submit an association file informing about the bucket–metabolite correspondence along with the dataset. This information will be used for all the following statistical approaches, automatically merging buckets associated with a same metabolite.

### 3.3. Novelty of the Study

The detailed approach used for our representative spectrum interpretation stands out from most of the usual NMR-based metabolomic studies in ecotoxicology. By exploiting the vast analytical potential of NMR spectrometry and associating several homonuclear and heteronuclear acquisition sequences (JRES, COSY, TOCSY, HSQC), we were able to provide a thoroughly argued description and annotation of a representative spectrum of *D. polymorpha* (Appendix A). Despite the interest of combining those approaches for the metabolomic spectra interpretation [30,31], the use of these 2D acquisition sequences to strengthen spectral interpretation remains scarce in NMR-based ecotoxicological metabolomic studies [55,56], especially on bivalve species [57,58]. However, even in those studies producing a higher confidence spectral interpretation, a validation of the signal assignment through standard metabolites spiking into the sample has rarely be performed.

As our main objective was to facilitate and increase the reliability of further work on *D. polymorpha* by other scientists, we make publicly available the annotated spectrum data file (see Appendix A). This spectrum data file can be opened on every NMR software package, and allows visualizing with high resolution the annotation of each peak from each detected signal directly on the spectrum. Figure 3 shows an example of what can be obtained by the reader on a random region. This annotated spectrum can be zoomed in on any region or signal of interest, and the Y axis can be adjusted according to the local signal intensities. This makes possible the direct comparison of our representative annotated spectrum with new spectra, allowing a rapid and easy annotation of signals detected in the new spectrum (Figure 3). Direct access to the annotated spectra is equally crucial to take into account with signal superposition when interpreting peaks’ intensity changes between several samples. To our knowledge, such annotated spectrum data files have never been released for any species relevant to ecotoxicological studies.

Overall, spectra interpretation and signal assignment are never really described and argued in ecotoxicological studies. In our study, we go to great length to describe all the arguments involved in the annotation process. In the same way, a reliably annotated spectra data file publication should be encouraged. This would allow both authors and readers to weigh and mitigate their biological interpretations and conclusions according to the reliability of the annotation of the metabolites of interest, while being careful in the consideration of potential signal superposition in the spectra. Improved accuracy of the biological interpretations from metabolomic approaches as here described would result. Thus, our work initiates a first move toward more conservative and more reliable reflective processes in the identification and interpretation of metabolomic changes in *D. polymorpha*, and perhaps in close-related species.

## 4. Materials and Methods 

### 4.1. Zebra Mussel Sampling

All mussels originated from the same population, located in the “Lac du Der-Chantecoq” (4°45’00’’ E, 48°34’00’’ N) (France, Appendix A) and were harvested in September 2017. The Dreissenidae species composition of this population was validated through a genetic marker [59] before sampling, confirming that only *D. polymorpha* occurs in this site. The animals were transferred and acclimated for two weeks in laboratory tanks with continuous aerated water and food supplementation.

To obtain a reference spectrum representative of the diversity of metabolic profiles in relationship to water quality (i.e., water conductivity), mussels were caged on 4 different sites located along the French part of the Meuse and Moselle river (Appendix A). These 4 sites were considered as reference/weakly impacted sites for the Meuse and Moselle rivers, and represented a panel of standard living conditions for mussels. Mussels were caged during the resting period, which is the privileged period for its use in biomonitoring, avoiding the important effect of the reproduction process on the organism’s metabolism. Mussels (20–25 mm) were then randomly distributed into 2-mm mesh polyethylene experimental cages (7 × 7 × 14 cm) (200 mussels per cage), and were left for 2 months (from 11 October 2017 to 12 December 2017) in 4 French sites located along the Moselle (1 site: 6°4’0.4303” E; 48°44’44.3620” N) and Meuse river (3 sites: 5°32’27.6180” E, 48°52’12.9349” N–4°44’28.1306” E, 49°48’58.8740” N–4°46’29.5799” E, 49°44’18.4070” N) (Appendix A). Physicochemical conditions of the 4 caging sites are given in the supporting information (Appendix A). Heavy metal and organic pollutants concentrations in soft tissues of mussels before and after being caged together with those used for metabolome analysis are provided in the supporting information (Appendix A). Randomly chosen individuals (*n* = 40) were collected per site (160 individuals, overall), deep-frozen in liquid nitrogen and stored at −80 °C until further processing. All relevant international, national and/or institutional guidelines for the care and use of *D. polymorpha* were followed during this study.

### 4.2. Sample Preparation

To ensure the representativeness of our reference sample, the 160 individuals collected after caging in 4 reference sites were used to create a unique homogenous representative sample (Mix sample). Sample preparation was optimized according to previously published work [4]. Soft tissues of frozen individuals were quickly removed on ice from their shell to avoid thawing, and whole soft bodies were placed in 2 mL Eppendorf tubes to be lyophilized for 24 h. This latter step ensured a maximal metabolome preservation during the following steps of sample processing. Lyophilized bodies were grinded using a Retsch MM400 ball mill (RETSCH GmbH, Haan, Germany), with two 3 mm diameter beads in each tube and 30 beats per minute for 5 min. The resulting powder from the set of 160 individuals was then pooled in a 50 mL plastic tube and mixed to form a homogenous whole-body dry powder (Mix sample). This powder was stored at −80 °C prior to metabolite extraction. Polar metabolite extractions were carried out from 50 mg of the Mix sample using a two-step methanol/chloroform/water method (final ratio 2:2:1.8) as reported hereafter. The Mix sample (50 mg) was homogenized with 13 µL/mg dry weight of ice-cold methanol (Prolabo HiPerSolv CHROMANORM super grad, VWR, Radnor, PA, USA) and 5.2 µL/mg cold ultrapure water (Merck KGaA, Darmstatd, Germany) into a 15 mL PYREX^®^ Glass Conical Centrifuge Tubes (Corning Inc., New York, NY, USA) and incubated on ice for 5 min after being vortexed for 30 s. Then, 13 µL/mg ice-cold chloroform (Prolabo rectapur, VWR, Radnor, PA, USA) and 6.5 µL/mg cold ultrapure water (ratio 2:1) were added and incubated on ice for 10 min after 30 s vortex homogenization. The two phases were separated by centrifugation at 2000× *g* for 5 min at 4 °C. The upper polar phase was carefully collected into 1.5 mL Eppendorf tubes and vacuum-dried by means of a Genevac 3HT-6 centrifugal evaporator system (SP Scientific, Warminster, PA, USA). The dried polar extracts were stored at −80 °C. This extraction process has been applied to 2 different 50 mg aliquots of the Mix sample. The first one allowed all 1D and 2D spectra acquisitions at 600 MHz, and the second one allowed the heteronuclear 2D experiments performed at 800 MHz (see Section 4.3). 

### 4.3. NMR Spectroscopy

The 1D and 2D ^1^H and ^13^C NMR representative spectra of the Mix sample were acquired on a 600 MHz Bruker AVANCE III spectrometer (Bruker, Biospin, Germany) equipped with a z-gradient 5 mm TCI cryoprobe at the Institute of Molecular Chemistry (ICMR) of Reims Champagne Ardenne University, France. One hour before the NMR analysis, the polar extract was suspended in 600 µL of 0.1M potassium phosphate buffer (pH 7.0) prepared in 99.9% D_2_O (Eurisotop, Saint-Aubin, France) containing 1 mM NaN_3_ (sodium azide, CAS26627-22-8) and 0.5 mM TSP (3-trimethylsilyl 2,2,3,3-d4 propionate, CAS 24493-21-8) as the internal standard. Vortexed samples were then transferred to 5 mm OD NMR tubes (Norell, Morganton, NC, USA) for spectra acquisition. Spectra were acquired and processed using the Bruker Topspin software, v3.2 and v4.0.6, respectively. All spectra were recorded at 298 K, using D_2_O (salted) for field locking with a 90° impulsion time of 11.5 µs (γ*B*_1_/2π = 21.7 kHz) and were referenced respective to the TSP signal at 0 ppm. The 1D ^1^H NMR spectrum was obtained using the noesygppr1d pulse sequence to supress the residual H_2_O resonance. The ^1^H spectrum recording consisted of 4 dummy scans (DS) followed by 128 acquired scans (NS), resulting in the collection of 128k real data points for a spectral width of 20 ppm. An exponential line-broadening filter function of 0.3 Hz width was applied before Fourier transformation, and the spectrum was manually phased on the zeroth order only. Careful adjustment of the FILCOR spectrometer delay parameter avoided the introduction of potential baseline deformation caused by a non-zero first order phase correction parameter. More detailed information on acquisition parameters and post-processing treatment is given in Appendix A.

Two-dimensional spectra were acquired on the same Mix sample in order to explore the complex structure of this representative ^1^H NMR spectrum. More detailed information on the acquisition parameters of these spectra is given in Appendix A. 

A homonuclear 2D ^1^H J-RESolved Spectroscopy (JRES) spectrum was obtained using the jresgpprqf sequence, with 8 dummy scans (DS) followed by 32 acquisition scans (NS) per t_1_ value. Spectral widths were set at 10 ppm in the acquisition dimension (F2) and to 40 Hz in the indirect dimension (*F*_1_). Overall, 64 FIDs were acquired, each of 8k real data points. The spectral sizes were set at 128 and 8k real data points along the indirect dimension (*F*_1_) and the direct one (*F*_2_), respectively. The spectrum was referenced on the TSP signal (0 ppm). The resulting 2D spectrum displays the signal multiplicity along *F*_1_ and the chemical shift along *F*_2_.

A homonuclear ^1^H-^1^H COrrelation Spectroscopy (COSY) spectrum was obtained using the cosygpprqf sequence, with 8 DS followed by 4 NS of 2048 real data points with a spectral width of 10 ppm in *F*_1_ and 4096 real data points with a spectral width of 10 ppm in *F*_2_. The spectrum was referenced on the TSP signal (0 ppm) in the two dimensions. The obtained 2D spectra allowed us to identify coupled ^1^H nuclei over three bonds in the ^1^H spectrum. Another homonuclear ^1^H-^1^H total correlation spectroscopy (TOCSY) spectrum was recorded using the dipsi2gpphpr pulse sequence, with 128 dummy scans (DS) followed by 8 acquisition scans (NS) of 2048 real data points with a spectral width of 10 ppm in *F*_1_ and 4096 real data points with a spectral width of 10 ppm in *F*_2_. This complementary homonuclear 2D experiment helps identifying the connected ^1^H spin systems and therefore facilitates the correct interpretation of the COSY spectrum.

The heteronuclear ^1^H-^13^C single-quantum correlation spectroscopy (HSQC) spectrum was obtained using the hsqcedetgpsisp2.2 acquisition sequence, with a 90° pulse duration of 13.5 µs (γ*B*_1_/2π = 18.5 kHz) for ^13^C. The spectrum was acquired with 256 DS and 8 NS of 1024 data points with a spectral width of 184 ppm in *F*_1_ and 2048 data points with a spectral width of 10 ppm in *F*_2_. The spectrum was referenced on the TSP signal (^1^H: 0 ppm; ^13^C: 184 ppm, folded along *F*_1_) on both dimensions. This spectrum allows to identify correlations between ^1^H and ^13^C chemical shifts for directly bound nuclei. 

Two additional heteronuclear HSQC spectra were acquired on a Bruker Avance III HD 800 MHz spectrometer equipped with a 5 mm quadruple resonance QCI-P (H/P−C/N/D) cryogenically cooled probe head at the MetaToul analytical facility (Toulouse Metabolomics & Fluxomics Facilities, Toulouse, France).

^31^P-^1^H HSQC-TOCSY and additional ^1^H-^13^C HSQC spectra were recorded on 50 mg of a new dry Mix extract. The dry extract was suspended in 200 µL D_2_O buffer described above, and transferred in a 3 mm OD NMR tube (Norell, Morganton, NC, USA). Spectra were acquired and processed using the Bruker Topspin 3.5pl7 software. The ^31^P-^1^H HSQC-TOCSY spectrum referenced on the TSP peak shift (^1^H: 0 ppm) on the second dimension. Referencing in *F*_1_ was performed based on the study of Gradwell et al. [60], with the adenosine mono-phosphate (AMP) ^31^P resonance set at δ_P_ -11.9. This spectrum allows one to identify correlations between each ^31^P resonance with those protons connected to the ^31^P nucleus through a three bond of a P-O-C-H motif (J ≈ 6Hz). Resonances of the other ^1^H nuclei that belong to the same spin system as the ^1^H nucleus within the already mentioned substructure are equally observed through a homonuclear proton 180° pulse train during the ^1^H, ^31^P INEPT transfer [60]. This spectrum, when recorded on glucose-1-phosphate, for example, would correlate the signal of the ^31^P nucleus to the one of the ^1^H nucleus at position 1. Correlations of the same ^31^P signal with other protons at positions 2, 3 and 4 can be equally observed, with decreasing intensities as in a regular TOCSY experiment. Detailed information on the acquisition parameters of these two spectra is given in Appendix A.

### 4.4. Metabolite Identification

The analysis of the ^1^H NMR spectrum of the Mix sample was facilitated by the study of the corresponding homonuclear 2D J-resolved (JRES) spectrum in order to determine the chemical shifts and the coupling constants for each signal produced by all the metabolites of the sample. The JRES spectrum makes it possible to group together the peaks in the 1D spectrum that are associated with the same nucleus and hence the same signal. A code number was assigned to each signal to allow for a simple designation for further works. The 2D ^1^H-^1^H COSY spectrum correlates the chemical shifts of coupled nuclei, thereby helping the identification of those pairs of ^1^H signals which are due to pairs of nearby hydrogen atoms within a molecule. This information was completed by the 2D ^1^H-^1^H TOCSY spectrum, providing information on ^1^H signals over multiple relayed couplings. Correlations between ^1^H and ^13^C chemical shifts were determined by means of the HSQC spectrum at 600 MHz, revealing single bonds between carbon and hydrogen atoms. In the final stage of this work, they were validated using the additional HSQC spectrum recorded at 800 MHz. A putative assignment of a ^1^H signal can therefore be confirmed by the assignment of its bound ^13^C signal, according to the available databases. All the ^1^H signals correlated by the COSY and TOCSY spectra were grouped together, and the chemical shift of all groups of signals was compared to the reference spectra from the MetaboHub PeakForestα database (v2.0.3) [61], and the Human Metabolome Database (HMDB) [62]. Once a plausible correspondence was identified, the coherence of all the other information obtained about the concerned signals was assessed. Only the signals with more than two arguments coherent with the databases (most of the time chemical shift, J and at least one homonuclear or heteronuclear correlation) and without any contradictory arguments were assigned to a metabolite. 

Remaining groups of signals or well-defined singlets unmatched to any reference in the database were identified and named as “Unknown Mx”. Among those, phosphorylated metabolites were identified through a comparison with the correlations in the ^1^H-^31^P HSQC spectrum, thus providing further characterization of those metabolites.

Several signals in the ^1^H NMR spectrum lack an argument for identification (e.g., a singlet signal without any identified correlation), and are equally characterized by the absence of good matches or comparable information in the reference databases. Mix samples prepared in the same way as the initial Mix sample were then spiked with 24 reference standard metabolites (maximum 4 metabolites added per extract) and analyzed by ^1^H NMR to confirm the annotation of several peaks. The detailed list of spiked metabolites and spiking concentrations is given in Appendix A.

The scale presently used for the assignment confidence estimation was adapted from the one proposed by the Metabolomics Standards Initiative [31]. “Level 1: Identified compound” was given to a signal assignment based on metabolite spiking in the sample. “Level 2: Putatively Annotated Compound” category was nuanced to differentiate two sub-levels of peak assignment. “Level 2+” was given to peaks identified by the search in the reference spectrum databases, with more than one confirmation from the coherent information obtained from the 2D NMR experiments. “Level 2−” refers to an assignment based on a reference spectrum from a database with only one confirmation (coherent ^1^H chemical shift and coupling constant if relevant), or corresponding to a poorly referenced metabolite or metabolite class in the databases. “Level 3: Putatively Characterized Compound Class” designates unidentified correlated groups of ^1^H signals that can correspond to several metabolites of a same type. Level 2−, level 3 and other groups of unidentified correlating groups of ^1^H peaks were referenced as “Unknown Mx” (Appendix A).

## 5. Conclusions

As often argued, the NMR-based metabolomic approach is relatively fast, with a moderate sample preparation time [17,28]. The acquisition program developed in this study allowed 1D ^1^H spectra acquisition in 36 to 40 min. This approach offers a real interest in ecotoxicological studies, as it makes practical complex experimental designs with multiple conditions, such as multi-site field experiments or dose–response experiments. The informative potential of the obtained spectra will allow detecting profile alterations due to the tested conditions, and to make assumptions about the specifically altered physiological processes. By making publicly available the detailed descriptions and annotation of the whole-body metabolome of *D. polymorpha*, our work should provide a reliable and useful resource for facilitating and improving the metabolomic data analysis and interpretation of this species. It is obvious that according to environmental conditions and the individual physiological state, this global profile will change, and maybe some of the identified metabolites will become undetectable and others that were undetected in our reference spectrum will be present. However, the global shape of the spectra and signal sequence across the chemical shift will be preserved. Using our 1D ^1^H NMR annotated spectrum and the detailed description of peaks’ shapes and coordinates, this reference spectrum provides a reliable basis for a first quick spectral interpretation, and eventually orientation of further peak assignment efforts according to the specific experiment. This offers the potential to foster the use of metabolomics approaches resorting on *D. polymorpha* to decipher the metabolic processes associated with natural physiological changes (e.g., reproduction) and to define the relevant biomarkers of the physiological process impairments. This is particularly true in large-scale multi-site ecotoxicological studies, such as comparative approaches using organisms from a same population caged on different sites along a contamination gradient to study the toxicity of water bodies or the impact of an effluent. Combined with other approaches such as mussel behavior analyses [63] and physiological stress biomarkers [6,7,8,9], this could greatly increase the understanding of water contaminants’ impacts on freshwater mussels. However, despite the interest of analyzing the metabolomic response of zebra mussels to ambient environmental conditions, this species is considered a major threat to native freshwater mussel species (e.g., [64]), and therefore should not be intentionally introduced into areas where they currently do not occur.

## Figures and Tables

**Figure 1 metabolites-10-00256-f001:**
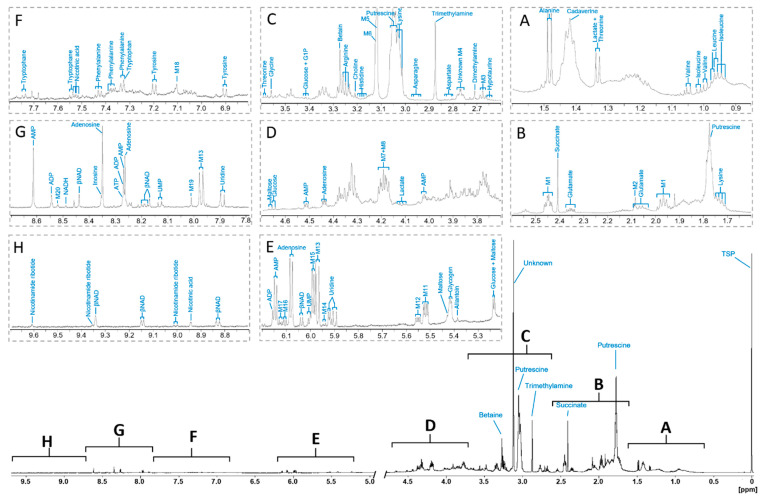
Representative ^1^H NMR spectrum of *Dreissena polymorpha* whole-body polar extract. Portions of the spectrum (**A**–**E**) are displayed and scaled according to the highest peak to improve readability. Only the most identifiable signals on the spectrum are labelled by their corresponding metabolite chemical name. Signals corresponding to unknown signals with ^1^H, ^13^C or ^31^P correlations are identified as M1 to M20. Detailed list of all identified signals in this spectrum and corresponding metabolites is provided as a Appendix A. The detailed list of identified metabolites is provided in Table 1. TSP (3-trimethylsilyl 2,2,3,3-d4 propionate): internal reference (0 ppm).

**Figure 2 metabolites-10-00256-f002:**
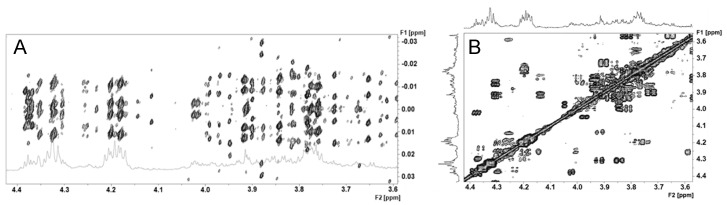
JRES ^1^H (**A**) and correlation spectroscopy (COSY) ^1^H-^1^H (**B**) NMR spectra in the 3.6 to 4.4 ppm region of the *Dreissena polymorpha* whole body. A high number of complex peak clusters with numerous correlated resonances is superposed in this region, making it difficult to interpret.

**Figure 3 metabolites-10-00256-f003:**
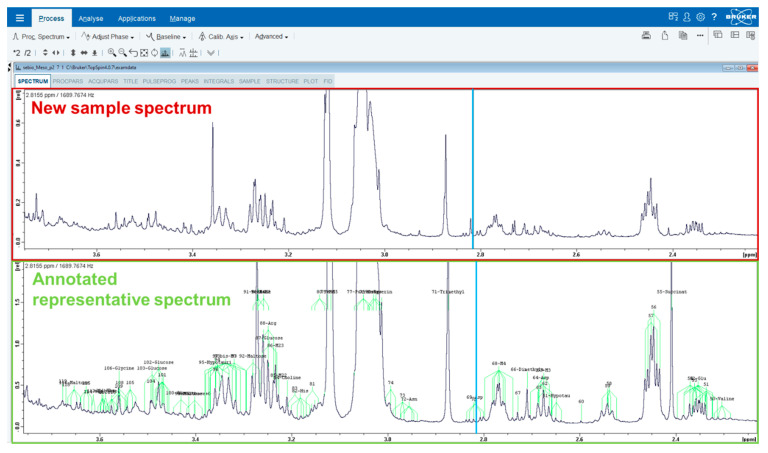
Screen capture exemplifying the comparison of our annotated representative spectrum (green frame) and a new one (red frame) on Topspin 4.0.7. The annotated spectrum (available online on Zenodo, see Appendix A section) can be zoomed in on a specific region of interest, and the y axis can be adjusted according to the signal intensity in the chosen region. Comparison of the representative spectrum to a new one allows easy signal identification thanks to the cursor (blue line) and global spectra profile visual comparison.

**Table 1 metabolites-10-00256-t001:** List of metabolites identified in the ^1^H NMR representative whole-body spectrum of *Dreissena polymorpha*, and their respective chemical shift and multiplicity (between brackets; s: singlet, d: doublet, t: triplet, q: quadruplet, dd: doublet of doublets, qd: quadruplet of doublets, m: multiplet, br: broad peak). If clearly identified on the J-resolved (JRES) spectrum, the number of peaks of a multiplet signal is specified. Number of identified peaks of the multiplet is given between brackets when the signal shape could not be confirmed by the JRES spectrum. Confidence levels: 1: assignment based on compound spiking in the sample; 2+: assignment based on a reference spectrum from a database, with several confirmations from 2D NMR experiments. 2−: assignment based on a reference spectrum from a database, with only one confirmation from 2D NMR experiments, or corresponding to poorly referenced spectra in databases. ▸Spiked metabolites. NMN: nicotinamide ribotide.

Metabolite	^1^H Chemical Shift (ppm)	Confidence Level
*Proteinogenic amino acids*		
Alanine	1.483 (d), 3.783 (q)	2+
Arginine	1.924 (m8), 3.249 (t), 3.776 (dd)	2+
▸Asparagine	2.956 (dd)	1
Aspartate	2.681 (q), 2.818 (dd), 3.899 (dd)	2+
Glutamate	2.061 (m6), 2.099 (m6), 2.136 (m12), 2.354 (m8), 3.760 (dd)	2+
▸Glycine	3.56 (s)	1
▸Histidine	3.183 (3), 7.965 (s)	1
Isoleucine	0.944 (t), 1.016 (d)	2+
Leucine	0.959 (d), 0.972 (d)	2+
▸Lysine	1.482 (m8), 1.729 (m4), 3.030 (t)	1
▸Phenylalanine	7.331 (d), 7.378 (m3), 7.418 (s), 7.430 (s), 7.442 (s)	1
Threonine	1.333 (d), 3.588 (d)	2+
▸Tryptophan	7.203 (t), 7.87 (t), 7.544 (d), 7.740 (d)	1
Tyrosine	6.902 (2), 7.196 (2)	2+
Valine	0.994 (d), 1.047 (d), 2.304 (dd), 3.613 (d)	2+
*Amine and amide compounds and osmolites*		
▸Choline	3.210 (s)	1
Phosphocholine	3.227 (s)	2−
Glycerophosphocholine	3.233 (s)	2−
Cadaverine	1.419 (m5), 1.733 (m6), 3.024 (t)	2+
Putrescine	1.775 (m9), 3.052 (m5)	2+
Trimethylamine	2.874 (s)	2+
Dimethylamine	2.709 (s)	2+
Betaine	3.272 (s), 3.913 (s)	2+
Hypotaurine	2.650 (t), 3.357 (t)	2+
Allantoin	5.387 (s)	2−
*Carboxylic acids*		
Succinic acid	2.409 (s)	2+
Lactic acid	1.331 (d), 4.114 (dd)	2+
Nicotinic acid	7.525 (qd), 8.253 (m8), 8.941 (s)	2+
γ-aminobutyric acid	1.967 (m5), 2.446 (t), 2.452 (t)	2−
Nucleotides–Nucleosides		
▸Adenosine	3.842 (dd), 3.918 (dd), 4.305 (dd), 4.439 (dd), 6.083 (d), 8.263 (s), 8.350 (s)	1
Inosine	8.239 (s), 8.355 (s)	2+
Uridine	4.369 (t), 5.900 (d), 5.924 (d), 7.889 (d)	2+
▸ATP	4.413 (m), 8.273 (s)	1
▸ADP	6.156 (d), 8.271 (s), 8.545 (s)	1
▸AMP	4.017 (d), 4.025 (d), 4.376 (m), 4.415 (qd), 6.147 (d), 8.268 (s), 8.615 (s)	1
▸UMP	6.001 (m3), 8.130 (d)	1
NMN	8.997 - 9.113 (d), 9.354 (s), 9.606 (s)	2−
Coenzymes		
▸βNAD	4.428 (m4), 4.489 (m4), 4.514 (qd), 4.546 (m), 6.040 (d), 8.169 (s), 8.118 (dd), 8.439 (s), 8.828 (dd), 9.145 (d), 9.340 (s)	1
▸NADH	8.490 (s)	1
Carbohydrates		
▸Glucose	(3.246 (d), 3.402 (dd), 3.430 (t), 3.477 (t), 3.491 (t), 3.727 (t), 4.650 (d), 5.237 (d)	1
▸Maltose	3.281 (t), 4.414 (s), 3.596 (2), 3.653 (dd), 3.698 (d), 3.725 (m6), 4.663 (d), 5.237 (d), 5.430 (2)	1
▸Glycogen	1.187 (t), 5.414 (br), 5.418 (br)	1

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
