# Peer review of "The Zebra Mussel (Dreissena polymorpha) as a Model Organism for Ecotoxicological Studies: A Prior 1H NMR Spectrum Interpretation of a Whole Body Extract for Metabolism Monitoring"

_metabolites, 2020, doi:10.3390/metabo10060256_

Round 1
Reviewer 1 Report
This work reports on the NMR based metabolomic approach on a lipid-free extract of zebra mussel (Dreissena polymorpha) to investigate its informative potential. In my opinion this report could represent a preliminary study based on the NMR interpretation and characterization of polar metabolites from a pool of whole body D. polymorpha powder. I appreciate that the characterization were performed through the analysis of the corresponding 2D homonuclear and heteronuclear NMR spectra, also 31P, and comparison with commercial compounds. However, I think that this work was built only on this redundant speculation. I suggest improving this research with MVA analyses to evaluate a first feasibility of these studies in the explorations of the impact of environmental changes on zebra mussel physiological state.
Author Response
This work reports on the NMR based metabolomic approach on a lipid-free extract of zebra mussel (Dreissena polymorpha) to investigate its informative potential. In my opinion this report could represent a preliminary study based on the NMR interpretation and characterization of polar metabolites from a pool of whole body D. polymorpha powder. I appreciate that the characterization were performed through the analysis of the corresponding 2D homonuclear and heteronuclear NMR spectra, also 31P, and comparison with commercial compounds. However, I think that this work was built only on this redundant speculation. I suggest improving this research with MVA analyses to evaluate a first feasibility of these studies in the explorations of the impact of environmental changes on zebra mussel physiological state.
The practical feasibility of metabolomic approaches in the explorations of the impact of environmental changes on mussels’ physiological state has already been demonstrated in zebra mussel by Wanatabe et al. (2015) and several researchers in other mussel species, as explained in the introduction of our article (L67-70). The objective of our work was not to re-demonstrate the feasibility of this approach on zebra mussel, but to accurately define the informative potential of zebra mussel 1H NMR spectrum and to provide a way to allow accurate interpretation of spectra changes observed in comparative studies on this species. Indeed, as stated in the introduction of our article, NMR based metabolomic approaches are too often based on putative assignment to specific metabolites (L81-91), potentially leading to inaccurate interpretations.
Reviewer 2 Report
The manuscript aims to provide a valid annotated NMR spectrum representative of the metabolic profile of the zebra mussels for assessing the informative potential of NMR-based metabolomics approach and to provide a reliable basis for metabolome analysis of this model species.
Overall, the topic of the present paper is of high interest and utility for the scientific community, and confirms NMR-based metabolomics as a powerful tool of high informative potential in the field of ecotoxicology. The manuscript is well structured and written, and discussion is good and well referenced. However, some revisions are needed in order to improve the quality of the manuscript.
In the Abstract, lines 18-20, revise the English grammar, mainly in the second part of the sentence.
Lines 59-68, for those not familiar with the potential of metabolomics, mainly in the field of aquatic ecology and ecotoxicology, it should be stated about the suitability of metabolomics to be applied in a wide range of aquatic organisms, and not only in marine bivalves, as well as its potential to successfully address several environmental questions, as recently reported in a Review on NMR-based metabolomics applied on aquatic organisms (Cappello 2020, eMagRes 9, 81-100, doi.org/10.1002/9780470034590.emrstm1604).
Lines 107-117, provide the country of the sampling site and of all the sites where zebra mussels were caged, together with a map depicting the study area with all sites clearly indicated. Also, provide more information on the type of pollution present in the caging sites, with references to previous studies conducted in this area with data on the level of contaminants present. Moreover, it is not clear why authors selected all caging sites as polluted and did not chose also a reference site not contaminated where to cage mussels. This needs to be clarified by authors also in the text of the manuscript.
Lines 123-144, it is not clear why authors collected 160 individuals from four different sites if at the end they did only a unique pool with all samples, if this is correct. This must be clearly specified. Also, clearly state how many samples from this pool were analysed by NMR to achieve the findings reported in this manuscript.
Line 253, unfortunately, Figure 1 is not present in the manuscript and must be provided. However, the list of metabolites assigned in the spectrum (mainly if this is clearly indicated in the spectrum itself) should be provided also in the figure caption to facilitate the reading of the spectrum, thus to avoiding readers to referring exclusively to the Supplementary file.
Lines 268-273, since in Figure 1 was not provided during submission, I wonder if all the metabolites listed in Table 1 are also clearly indicated in the NMR spectrum. This would greatly help the reader, particularly those not familiar with this approach, and will surely represent an additional value to this work, which aim is also that to provide a spectrum to be used as a reference for future studies.
Lines 454-457, it must be reported that the relaxant neurotransmitter GABA plays also an important role in the adaptive strategies elicited by organisms during anoxic or hypoxic conditions, as recently documented in pen shells by Cappello et al. 2019, The European Zoological Journal 86, 333-342, https://doi.org/10.1080/24750263.2019.1673492.
Lines 463-66, according to this statement and potential of NMR, together with identification provide also quantitative data on the concentrations of metabolites detected in the spectrum, or at least of all those metabolites listed in Table 1.
Author Response
Reviewer 2
- In the Abstract, lines 18-20, revise the English grammar, mainly in the second part of the sentence.
The initial sentence “A nuclear magnetic resonance (NMR) based metabolomic approach was developed in this context, which required the prior characterization of a reference 1H NMR spectrum of a lipid-free zebra mussel extract and to investigate its informative potential.” Was replaced as follow (line 18-20): “A nuclear magnetic resonance (NMR) based metabolomic approach was developed on this species. The investigation of its informative potential required the prior interpretation of a reference 1H NMR spectrum of a lipid-free zebra mussel extract”
- Lines 59-68, for those not familiar with the potential of metabolomics, mainly in the field of aquatic ecology and ecotoxicology, it should be stated about the suitability of metabolomics to be applied in a wide range of aquatic organisms, and not only in marine bivalves, as well as its potential to successfully address several environmental questions, as recently reported in a Review on NMR-based metabolomics applied on aquatic organisms (Cappello 2020, eMagRes 9, 81-100, doi.org/10.1002/9780470034590.emrstm1604).
We agree with you, and we already mentioned this potential at line 63-65 “In addition, metabolomic approaches have the great advantage of being applicable to any species without any prior knowledge of its genome.”. According to this comment, we emphasized this point in the sentence L67-69: “Metabolomics has been involved in several studies related to the impact of contaminants on numerous aquatic organisms, including marine bivalve species…”.
- Lines 107-117, provide the country of the sampling site and of all the sites where zebra mussels were caged, together with a map depicting the study area with all sites clearly indicated. Also, provide more information on the type of pollution present in the caging sites, with references to previous studies conducted in this area with data on the level of contaminants present. Moreover, it is not clear why authors selected all caging sites as polluted and did not chose also a reference site not contaminated where to cage mussels. This needs to be clarified by authors also in the text of the manuscript.
The sampling and caging sites are located in France. We added this information in section 2.1 (L108, L114, L120), and we additionally provided a map of the sites of sampling and caging in supporting information (Figure S1).
The physicochemical parameters of the caging sites during the 2 months of caging have been added to our article in the “supporting information” files (Table S1). Heavy metal and organic pollutants concentrations in soft tissues of mussels caged together with those used for metabolome analysis have been added to our article in the supporting information file (Table S2 and 3).
The 4 caging sites are indeed considered as “reference sites” for Meuse and Moselle rivers. However, considering that there is no river in France (and in the world) that can be safely characterized as absolutely not impacted or contaminated by human activities and pollutants of any kind, we used the term “moderately impacted/contaminated” to qualify these sites. We agree that this may be confusing for the reader, and we clarified this point in the article as follow: “To obtain a reference spectrum representative of the diversity of metabolic profiles in relationship to water quality, mussels were caged on 4 different sites located along the French part of the Meuse and Moselle river (Figure S1). These 4 sites were considered as reference/weakly impacted sites for the Meuse and Moselle rivers, and represented a panel of standard living conditions for mussels” (L 113-116).
- Lines 123-144, it is not clear why authors collected 160 individuals from four different sites if at the end they did only a unique pool with all samples, if this is correct. This must be clearly specified. Also, clearly state how many samples from this pool were analysed by NMR to achieve the findings reported in this manuscript.
As stated in the previous section of the “Material and methods” (2.1), we used individuals caged in four different sites to insure a diversity of metabolic profiles in relationship to water quality in our reference spectrum. We stressed this point by rearranging the 2.1 section (L 113-116). In addition, we clarified this at the beginning of 2.2 section (L131-133) “To ensure the representativeness of our reference sample, the 160 individuals collected after caging in 4 reference sites were used to create a unique homogenous representative sample (Mix sample)”.
NMR experiments at 600 MHz and 800 MHz were performed on two separate Mix sample extracts. This has been specified in the 2.2 section of the article (L 150-153) “This extraction process has been applied to 2 different 50 mg aliquot of the Mix sample. The first one allowed all 1D and 2D spectra acquisition at 600MHz, and the second one allowed the heteronuclear 2D experiment performed at 800 MHz (see section 2.3)”
- Line 253, unfortunately, Figure 1 is not present in the manuscript and must be provided. However, the list of metabolites assigned in the spectrum (mainly if this is clearly indicated in the spectrum itself) should be provided also in the figure caption to facilitate the reading of the spectrum, thus to avoiding readers to referring exclusively to the Supplementary file.
Unfortunately, Fig. 1 seems to have been lost during the submission process. We re-inserted figure 1 (page 7 of 18). Metabolite names are directly indicated on the spectra sections presented in Fig. 1. The list of identified metabolites (and the chemical shift of the corresponding signals) are provided in table 1 of the article (page 8 of 18). We modified Fig 1 caption to clarify this (line 293-294).
- Lines 268-273, since in Figure 1 was not provided during submission, I wonder if all the metabolites listed in Table 1 are also clearly indicated in the NMR spectrum. This would greatly help the reader, particularly those not familiar with this approach, and will surely represent an additional value to this work, which aim is also that to provide a spectrum to be used as a reference for future studies.
Yes, all the metabolites listed in Table 1 are clearly indicated on Fig 1, at least for their most identifiable signal. Interested readers can also visualize the detailed assignation of all the detected signals on NMR software (eg Topspin) the 1H reference spectrum provided online on Zenodo (not available until publication), and you can download it at this URL: https://filesender.renater.fr/?s=download&token=f0cd25d8-2f69-4e33-98dd-e0c045f4b2e2
- Lines 454-457, it must be reported that the relaxant neurotransmitter GABA plays also an important role in the adaptive strategies elicited by organisms during anoxic or hypoxic conditions, as recently documented in pen shells by Cappello et al. 2019, The European Zoological Journal 86, 333-342, https://doi.org/10.1080/24750263.2019.1673492.
Thanks for this additional information that we missed. We included it and the corresponding reference in the manuscript (L471-474). “. It is well described for its inhibitory neurotransmitter activity, but is also involved in immune response homeostasis in pacific oyster [50] and in adaptive strategies to anoxic or hypoxic conditions in the pen shell Atrina rigida [51]. Its levels decrease in the freshwater mussel Elliptio complanata upon exposure to neurochemicals [52]. Its abundance could thus provide information about the state of GABAergic signaling, with potential implication for the immune response state and oxygen supply in the organism.”
- Lines 463-66, according to this statement and potential of NMR, together with identification provide also quantitative data on the concentrations of metabolites detected in the spectrum, or at least of all those metabolites listed in Table 1.
We assume that this comment refers to the terms “D. polymorpha metabolic studies allows semi-quantitative identification of key metabolic compounds”. By “semi-quantitative” we mean that signal intensities in the NMR spectra for a given molecule are directly linked to the concentration of the corresponding metabolite in the sample. The interest of our work therefore is more focused on the identification of the detected metabolites than on the quantification of their abundance. The present study does not pretend to benchmark concentrations for a given set of metabolites, as it will highly depend on environmental conditions or reproductive state for example. In that sense, the quantification of metabolite concentrations in our reference spectrum seems of lesser importance to us than the identification of the largest set of signals as possible. However, as we provide all our data and spectra in free access with this article, interested readers will be able to quantify the metabolites themselves using the internal standard signal (TSP) if they have a reason to do so. (those data are accessible at this URL: https://filesender.renater.fr/?s=download&token=f0cd25d8-2f69-4e33-98dd-e0c045f4b2e2)
Reviewer 3 Report
This is a well designed, conducted and presented study.
I only have some minor comments to this quite enjoyable read:
- please use 'D. polymorpha' after the first mention;
- Indicate that a collection site is in France;
- Provide a separate section 'Conclusions' after the Discussion.
Additionally, a file available to me lacked figure 1. Please upload it.
Author Response
Reviewer 3
- please use 'D. polymorpha' after the first mention;
We proceeded to this correction throughout the text (L73, 101, 133, 274, 277)
- Indicate that a collection site is in France;
We added this information in section 2.1 (L108, L114, L120), and we additionally provided a map of the sites of sampling and caging in supplementary information (Figure S1).
- Provide a separate section 'Conclusions' after the Discussion.
Following this advice, the 4.3 section of the Discussion part has been transferred in a new Conclusions section (5., L530)
- Additionally, a file available to me lacked figure 1. Please upload it.
Unfortunately, Fig. 1 seems to have been lost during the submission process. We re-inserted figure 1 (page 7 of 18).
Reviewer 4 Report
This paper provides NMR Spectra from whole body extracts for metabolism monitoring in the freshwater mussel Dreissena polymorpha as a basis for future assessments of impacts of environmental stressors and contaminants on the species. The choice of zebra mussel is well-justified since this is a globally invasive and thus wide-spread species. The paper is very well written and easy to follow, it is clearly state-of-the-art and all methodological aspects are well laid out, the results are clear and the discussion is to the point. I thus only have a few suggestions that may help to further improve the paper:
- Quite often, zebra mussel (Dreissena polymorpha) can get confused with quagga mussel (Dreissena bugenis) and both species can co-occur in the same habitats. I think it is thus useful to mention this risk and recommend genetic species verification. This could potentially also be an additional explanation for the differences in the metabolome observed with previous screenings by others mentioned in the discussion. Maybe the authors can add a phrase such as: “Due to shell morphological similarities and plasticity zebra mussel (Dreissena polymorpha) and its sister taxon quagga mussel (Dreissena bugenis), a genetic species verification is recommended (Beggel et al. 2015).
- The authors make a strong point in stressing the potential of metabolomics analyses in mussels. It would even become stronger if it was stressed that a combination of this methodology with other recently established ecotoxicological endpoints such as mussel behavior analyses (Hartmann et al. 2016) and physiological stress indicators (Beggel et al. 2017) can greatly increase our understanding of contaminant impacts on freshwater mussels. This would bridge the gap to other approaches of using mussels in ecotoxicology.
- Due to the usefulness of the paper, the approach is likely to also become applied by readers of the article. This comes with the danger that people may introduce zebra mussels to ecosystems where they are not yet present and therefore I find it important to at least include a brief warning that this globally invasive species is a major threat to freshwater biodiversity that must not be spread. For instance, include a phrase such as: “Despite of the usefulness of analyzing metabolomes from zebra mussels to ambient environmental conditions, they are considered a major threat to native mussel species (e.g. Ozgo et al., 2020) that should not be introduced into areas where they currently do not occur.
References:
Beggel S, Hinzmann M, Machado J, Geist J (2017) Combined Impact of Acute Exposure to Ammonia and Temperature Stress on the Freshwater Mussel Unio pictorum. Water 9; 455. DOI: 10.3390/w9070455
Beggel S, Cerwenka AF, Brandner J, Geist J (2015) Shell morphological versus genetic identification of quagga mussel (Dreissena bugensis) and zebra mussel (Dreissena polymorpha). Aquatic Invasions 10; 93-99
Hartmann J, Beggel S, Auerswald K, Stoeckle BC, Geist J (2016) Establishing mussel behavior as a biomarker in ecotoxicology. Aquatic Toxicology 170; 279-288
Ożgo, M, UrbaÅ„ska, M, Hoos, P, et al. (2020) Invasive zebra mussel (Dreissena polymorpha) threatens an exceptionally large population of the depressed river mussel (Pseudanodonta complanata) in a postglacial lake. Ecol Evol. 2020; 00: 1– 10. https://doi.org/10.1002/ece3.6243
Author Response
Reviewer 4
- Quite often, zebra mussel (Dreissena polymorpha) can get confused with quagga mussel (Dreissena bugenis) and both species can co-occur in the same habitats. I think it is thus useful to mention this risk and recommend genetic species verification. This could potentially also be an additional explanation for the differences in the metabolome observed with previous screenings by others mentioned in the discussion. Maybe the authors can add a phrase such as: “Due to shell morphological similarities and plasticity zebra mussel (Dreissena polymorpha) and its sister taxon quagga mussel (Dreissena bugenis), a genetic species verification is recommended (Beggel et al. 2015).
That’s a good remark. The population living in our sampling site (Lac du Der-Chantecoq) has been tested in 2017 thanks to the genetic marker published in Kerambrun et al 2018 (https://doi.org/10.1016/j.ecoenv.2018.02.051). Those results have not been published, however, it revealed that Lac du Der-Chantecoq population is only composed of D. polymorpha. We added a sentence in this sense in the Materials and Methods section (L108-110: " Dreissenidea species composition of this population has been validated through genetic marker [35] before sampling, confirming that only D. polymorpha occur in this site.”)
- The authors make a strong point in stressing the potential of metabolomics analyses in mussels. It would even become stronger if it was stressed that a combination of this methodology with other recently established ecotoxicological endpoints such as mussel behavior analyses (Hartmann et al. 2016) and physiological stress indicators (Beggel et al. 2017) can greatly increase our understanding of contaminant impacts on freshwater mussels. This would bridge the gap to other approaches of using mussels in ecotoxicology.
Thanks for this relevant suggestion, that allowed us to improve the “Conclusions” section of the article (L 542-544) “Combined to other approaches such as mussel behavior analyses [57] and physiological stress biomarkers [6,7,8,9], this could greatly increase the understanding of water contaminants impacts on freshwater mussels.”
- Due to the usefulness of the paper, the approach is likely to also become applied by readers of the article. This comes with the danger that people may introduce zebra mussels to ecosystems where they are not yet present and therefore I find it important to at least include a brief warning that this globally invasive species is a major threat to freshwater biodiversity that must not be spread. For instance, include a phrase such as: “Despite of the usefulness of analyzing metabolomes from zebra mussels to ambient environmental conditions, they are considered a major threat to native mussel species (e.g. Ozgo et al., 2020) that should not be introduced into areas where they currently do not occur.
This remark is highly relevant, and we added a sentence in that way at the end of the manuscript (L544-547) “However, despite the interest of analyzing metabolomic response of zebra mussels to ambient environmental conditions, this species is considered a major threat to native freshwater mussel species (e.g. [58]), and therefore should not be intentionally introduced into areas where they currently do not occur.”

Reviewer 5 Report
In this manuscript, the AA present an accurate interpretation of the proton NMR spectrum of the whole body extract of a freshwater mussel. The paper is therefore interesting and approaches chosen are appropriate. Some grammar inaccuracies are present along the text and must be reviewed. Moreover, some useful information are missing, mainly a description of the study area where mussels were collected, and also the NMR spectrum of freshwater mussels, and must be therefore provided by AA.
In the abstract and throughout the text, delete any space between number and percentage.
Check the references, as in line 83 the number 26 is present twice. Revise it.
In Materials and Methods section, a figure with the study area and all sites selected is missing and must be provided, as well as indication of the country where the sampling of freshwater mussels was conducted.
Information of the pollution status and data on the pollutants present, as well as their levels in the sampling sites, must be provided by AA together with a description of the study area.
How many samples were analyzed for NMR analysis? Did the AA compare spectra from mussels collected from the different sites? This seems unclear in the Materials and Methods section.
Several sentences along the text start with a number, and this should be avoided. Modify it, accordingly. An accurate revision throughout the text is needed, including Discussion section.
Some grammar inaccuracies are present at line 203, “can equally observed, be it...”
Where is Figure 1? Unfortunately, it was not provided even if the novelty of the paper and all the results reported are referred to the NMR spectrum of freshwater mussels. However, from the figure caption it seems that peaks in the spectrum were not assigned with metabolites but it is very important to include and clearly show the resonances of metabolites in the spectrum.
At line 398: were or where? Check and correct it.
For a correct and accurate interpretation of the spectrum, a quantitative analysis of all metabolites detected in the freshwater mussels should be provided. This would notably enrich the quality of the paper highlighting a further benefit from applying a NMR-based metabolomics approach. Moreover, the spectrum provided could be better used as a reference for future works conducted using this freshwater mussel as a model species.
Suggestion of the referee:
- Acceptable for publication with minor revisions.
Author Response
Reviewer 5
- In the abstract and throughout the text, delete any space between number and percentage.
This has been done throughout the text (L25, 278, 402, 404)
- Check the references, as in line 83 the number 26 is present twice. Revise it.
References were checked throughout the text.
- In Materials and Methods section, a figure with the study area and all sites selected is missing and must be provided, as well as indication of the country where the sampling of freshwater mussels was conducted.
The sampling was performed in France. We added this information in section 2.1 (L108, L114, L120), and we additionally provided a map of the sites of sampling and caging in supporting information (Figure S1).
- Information of the pollution status and data on the pollutants present, as well as their levels in the sampling sites, must be provided by AA together with a description of the study area.
The 4 caging sites are considered as reference sites for Meuse and Moselle rivers. However, considering that there is no river in France (and in the world) that can be safely characterized as absolutely not impacted or contaminated by human activities and pollutants of any kind, we used the term “moderately impacted/contaminated” to qualify these sites. As pointed here, this may be confusing for the reader, and we clarified this point in the article as follow: “To obtain a reference spectrum representative of the diversity of metabolic profiles in relationship to water quality, mussels were caged on 4 different sites located along the French part of the Meuse and Moselle river (Figure S1). These 4 sites were considered as reference sites for the Meuse and Moselle rivers, and represented a panel of standard living conditions for mussels” (L 113-116).
In addition, the physicochemical parameters of the caging sites during the 2 months of caging have been added to our article in the supporting information file (Table S1). Heavy metal and organic pollutants concentrations in soft tissues of mussels caged together with those used for metabolome analysis have been added to our article in the supporting information file (Table S2 and 3).
- How many samples were analyzed for NMR analysis? Did the AA compare spectra from mussels collected from the different sites? This seems unclear in the Materials and Methods section.
We agree that this point was quite unclear in the first version of our manuscript. Our objective was to obtain a reference spectrum representative of the diversity of metabolic profiles under a panel of standard living conditions. Then, the 160 collected individuals were pooled in a unique representative sample (Mix sample). This was clarified at the beginning of 2.2 section (L131-133) “To ensure the representativeness of our reference sample, the 160 individuals collected after caging in 4 reference sites were used to create a unique homogenous representative sample (Mix sample)”. Then, NMR experiments at 600 MHz and 800 MHz were performed on two separate extracts of this Mix sample. This has been specified in the 2.2 section (L 150-153) “This extraction process has been applied to 2 different 50 mg aliquot of the Mix sample. The first one allowed all 1D and 2D spectra acquisition at 600MHz, and the second one allowed the heteronuclear 2D experiments performed at 800 MHz (see section 2.3)”.
- Several sentences along the text start with a number, and this should be avoided. Modify it, accordingly. An accurate revision throughout the text is needed, including Discussion section.
This kind of mistakes have been revised throughout the text (L126; 142; 145; L277-278; L404; L493)
- Some grammar inaccuracies are present at line 203, “can equally observed, be it...”
The mistake has been corrected (L216)
- Where is Figure 1? Unfortunately, it was not provided even if the novelty of the paper and all the results reported are referred to the NMR spectrum of freshwater mussels. However, from the figure caption it seems that peaks in the spectrum were not assigned with metabolites but it is very important to include and clearly show the resonances of metabolites in the spectrum.
Unfortunately, Fig. 1 seems to have been lost during the submission process. We re-inserted figure 1 (page 7 of 18). Metabolite names are directly indicated on the spectra sections presented in Fig. 1.
- At line 398: were or where? Check and correct it.
The mistake has been corrected (L412)
- For a correct and accurate interpretation of the spectrum, a quantitative analysis of all metabolites detected in the freshwater mussels should be provided. This would notably enrich the quality of the paper highlighting a further benefit from applying a NMR-based metabolomics approach. Moreover, the spectrum provided could be better used as a reference for future works conducted using this freshwater mussel as a model species.
NMR experiments are semi-quantitative, meaning that signal intensities in the NMR spectra for a given molecule are directly linked to the concentration of the corresponding metabolite in the sample. However, ecotoxicological metabolomic studies are always based on differential signal intensities between individuals from different sites or condition. The interest of our work therefore is more focused on the identification of the detected metabolites than on the quantification of their abundance. The present study does not pretend to benchmark concentrations for a given set of metabolites, as it will highly depend on environmental conditions or reproductive state for example. In that sense, the quantification of metabolite concentrations in our reference spectrum seems of lesser importance to us than the identification of the largest set of signals as possible. However, as we provide all our data and spectra in free access with this article, interested readers will be able to quantify the metabolites themselves using the internal standard signal (TSP) if they have a reason to do so. (those data are accessible at this URL: https://filesender.renater.fr/?s=download&token=f0cd25d8-2f69-4e33-98dd-e0c045f4b2e2).